# Effects of a Lacto-Ovo-Vegetarian Diet on the Plasma Lipidome and Its Association with Atherosclerotic Burden in Patients with Coronary Artery Disease—A Randomized, Open-Label, Cross-over Study

**DOI:** 10.3390/nu12113586

**Published:** 2020-11-23

**Authors:** Demir Djekic, Lin Shi, Fredrik Calais, Frida Carlsson, Rikard Landberg, Tuulia Hyötyläinen, Ole Frøbert

**Affiliations:** 1Department of Cardiology, Faculty of Health, Örebro University Hospital, 701 85 Örebro, Sweden; demir.djekic@oru.se (D.D.); fredrik.calais@regionorebrolan.se (F.C.); ole.frobert@regionorebrolan.se (O.F.); 2School of Food Engineering and Nutritional Science, Shaanxi Normal University, 710061 Xi’an, China; 3Division of Food and Nutrition Science, Department of Biology and Biological Engineering, Chalmers University of Technology, 412 96 Gothenburg, Sweden; frittecarlssons@gmail.com (F.C.); rikard.landberg@chalmers.se (R.L.); 4Department of Public Health and Clinical Medicine, Umeå University, 901 87 Umeå, Sweden; 5Department of Chemistry, Örebro University, 701 82 Örebro, Sweden; tuulia.hyotylainen@oru.se

**Keywords:** coronary artery disease, randomized controlled trial, vegetarian diet, lipidomics

## Abstract

A vegetarian diet has been associated with a lower risk of coronary artery disease (CAD). Plasma triacylglycerols, ceramides, and phosphatidylcholines may improve prediction of recurrent coronary events. We sought to investigate effects of a lacto-ovo-vegetarian diet (VD) on plasma lipidome in CAD patients and simultaneously assess associations of plasma lipids with the extent of coronary atherosclerotic burden. We analyzed 214 plasma lipids within glycerolipid, sphingolipid, and sterol lipid classes using lipidomics from a randomized controlled, crossover trial comprising 31 CAD patients on standard medical therapy. Subjects completed a four-week intervention with VD and isocaloric meat diet (MD), separated by a four-week washout period. The VD increased levels of 11 triacylglycerols and lowered 7 triacylglycerols, 21 glycerophospholipids, cholesteryl ester (18:0), and ceramide (d18:1/16:0) compared with MD. VD increased triacylglycerols with long-chain polyunsaturated fatty acyls while decreased triacylglycerols with saturated fatty acyls, phosphatidylcholines, and sphingomyelins than MD. The Sullivan extent score (SES) exhibited on coronary angiograms were inversely associated with triacylglycerols with long-chain polyunsaturated fatty acyls. Phosphatidylcholines that were lower with VD were positively associated with SES and the total number of stenotic lesions. The VD favorably changed levels of several lipotoxic lipids that have previously been associated with increased risk of coronary events in CAD patients.

## 1. Introduction

Coronary artery disease (CAD) is a leading cause of morbidity and death worldwide [1]. A global change in diet characterized by a reduction in caloric intake, higher intake of plant-based food, and a reduced consumption of red and processed meat has the potential to prevent or delay the development of non-communicable diseases [2]. Epidemiological studies have shown that a lacto-ovo-vegetarian diet (VD) is associated with a 29% risk reduction in CAD mortality [3] and meta-analyses of randomized controlled trials have shown that such a diet favorably affects CAD risk factors including total cholesterol, low-density lipoprotein cholesterol (LDL-C), high-density lipoprotein cholesterol (HDL-C), blood pressure, and body weight [4,5]. Although blood triacylglycerol (TG) level has also been considered an independent risk factor for CAD, findings of randomized controlled trials regarding effects of a vegetarian diet on its concentration remain inconsistent [4,6,7].

Molecular lipid species, including glycerolipids, phospholipids, sphingolipids, and ceramides are involved in various biological functions including energy storage, serving as precursors for metabolic processes and as main components of cellular building blocks [8]. Perturbations in lipid metabolism can lead to lipotoxicity, generally defined as an increased concentration of detrimental lipids such as specific lipids with saturated fatty acyls and ceramides [9]. Indeed, dysregulation of lipid metabolism has often been reported in cardiovascular disease [10,11]. Recent large-scale lipidomics analyses of epidemiological studies indicate that ceramides and phosphatidylcholines may improve prediction of recurrent CAD events, independent of traditional CAD risk factors [10,11]. Dietary lipid overload, in addition to disturbed adipose tissue metabolism play major roles in accumulation of lipotoxic lipids [12].

We previously found that the extent of coronary atherosclerosis assessed by the Sullivan extent score in CAD patients was associated with extra-coronary artery disease, all-cause mortality, and poor prognosis following myocardial infarction [13]. It has been reported that the extent of coronary atherosclerosis is positively associated with levels of total cholesterol and LDL-C, and inversely associated with HDL-C. However, randomized controlled trials investigating effects of a vegetarian diet on the lipid profile or on lipid metabolic pathways are lacking. Such information may be of importance in identifying targets for CAD prevention and management using dietary approaches.

In a recent randomized open-labelled cross-over study, we reported that a 4-week VD improved levels of oxidized low-density lipoprotein cholesterol and cardiometabolic risk factors including body mass index (BMI), body weight, total cholesterol, and low-density lipoprotein cholesterol, compared with an isocaloric meat-containing diet (MD) [14]. The goal of the current exploratory study was to determine the effects of a VD on the plasma lipidome and to investigate associations between plasma lipids and CAD burden, assessed by coronary angiogram from a recent percutaneous coronary intervention (PCI).

## 2. Materials and Methods 

### 2.1. Study Participants

Subjects with CAD treated with PCI and on optimal medical therapy were recruited from patients at the Department of Cardiology, Örebro University Hospital, Sweden, from September 2017 through March 2018. The inclusion criteria were >18 years of age, stable CAD, PCI more than 30 days prior to inclusion in the study, and receiving optimal medical therapy including cholesterol-lowering drugs and aspirin. Exclusion criteria included age <18 years, unstable cardiovascular disease, PCI treatment in the 30 days prior to inclusion, inability to provide informed consent, already following a vegetarian or a vegan diet, vitamin B deficiency, known food allergy, previous surgery for obesity or gastric bypass surgery, or life expectancy <1 year. All participants provided oral and written informed consent. The primary outcome of this trial was to investigate difference in change in plasma oxidized LDL-C between diets, which has been published elsewhere [14]. The sample-size calculation was based on previous studies in which a VD or food supplements (nuts, soy-based cereal, cranberry juice) were shown to reduce oxidized LDL-C by 10% compared to no intervention and was based on an estimated 10% dropout rate. A total of 31 subjects were enrolled [14]. 

### 2.2. Study Design

The flowchart of the study is presented in Figure 1. The study design has been described previously [14]. In brief, the vegetarian diet in patients with ischemic heart disease (VERDI) trial was a prospective, randomized, open-labelled, cross-over study. Subjects consumed a four-week VD and a four-week isocaloric diet with daily meat consumption, separated by a four-week washout period. Participants were randomly allocated 1:1 to intervention sequences VD/MD or MD/VD. Data were obtained at baseline, following the first intervention, at the end of the washout period, and following the second intervention. The study is registered at ClinicalTrials.gov (NCT02942628) and the regional ethical review board in Uppsala, Sweden approved the study (Dnr 2016/456). Clinical Trial Registration: URL: https://www.clinicaltrials.gov; Unique identifier: NCT02942628.

### 2.3. Dietary Interventions and Compliance

Detailed information of diets has been described previously [14]. Subjects were provided with ready-made frozen lunches and dinners based on traditional Swedish recipes, produced and supplied by Dafgård, Källby, Sweden. The VD allowed intake of eggs and dairy products. We based the MD on the average meat consumption in Sweden of approximately 145 g/day. Subjects were responsible for weekly collection of the meals from the cardiology department. At the first study visit, subjects met with a research dietitian who provided instruction on strictly following the meal plans, which were energy-adjusted based on subject’s basal metabolic rate and physical activity level according to Henry’s equation. In addition to the two main meals provided, participants were asked to have breakfast and two light snacks that were selected from provided options every day. The diet plans conformed to the Nordic Nutrition Recommendations [15]. Nutrition calculation software (Dietist Net Pro; Kost och Näringsdata AB, Bromma, Sweden) was used to determine the nutrient composition of the diets. A three-day weighed food record showed both dietary interventions to be well-tolerated, and overall adherence based on self-reported diaries was 88% for both interventions. Detailed information has been presented elsewhere [14].

### 2.4. Measurements of Cardiovascular Risk Factors

Subject height was measured in centimeters at baseline. At the four study visits, the following values were assessed: blood pressure and heart rate with a digital automatic sphygmomanometer in the right arm after five minutes of seated rest; body weight in kilograms, with the participants dressed in light clothing without shoes; BMI determined as weight in kilograms divided by height in meters squared; total cholesterol (TC), LDL-C, HDL-C, and total triglycerides assessed using a dry chemistry standardized method. High sensitivity CRP (hs-CRP) was measured with two-site sandwich assays. HbA_1c_ was measured using high performance liquid chromatography [14]. Oxidized LDL-C was assessed using an enzyme-linked immunosorbent assay kit (Mercodia, Uppsala, Sweden). 

### 2.5. Assessment of Coronary Artery Disease Burden

We estimated the total CAD burden by eye balling visual analysis of coronary angiograms, performed a median of 219 days prior to baseline (interquartile range 146–348) (Figure 1). The coronary tree was divided into 15 segments as defined by the American Heart Association (Figure 2). In each segment, we assessed stenosis diameter, number of stenotic lesions, and longitudinal extent of atherosclerosis. The degree of stenosis in each segment was defined as the percentage of luminal narrowing in the most severely stenosed lesion of each coronary segment. A non-narrowed section approximately 5 mm proximal or distal to the stenotic lesion was used as reference to estimate percentage of diameter stenosis. A significant stenosis was defined as >50% narrowing. The number of stenotic lesions was counted in each segment. Any stenotic lesion extending twice the distance of normal lumen diameter was considered two lesions. A maximum number of three stenotic lesions was considered in each segment. We calculated the total number of stenotic lesions in all coronary arteries. The branches distal to a stenosis >50% were assessed before and after PCI. Branches estimated at <1 mm diameter were not included in the analysis.

The Sullivan extent score (SES) was used to classify the longitudinal extent of CAD [16]. Coronary artery disease was defined as irregularity of any vessel wall obstructing >20% of the total lumen and was estimated relative to the total length of the studied segment. An occluded lesion in any segment was defined as at least 50% longitudinal extent. The proportion of the longitudinal extent of each segment occupied by the lesion was multiplied by a factor representing the surface area of the studied segment relative to the entire coronary tree. The left main coronary artery accounted for 5%, the left anterior descending artery 35%, the left circumflex artery 30%, and right coronary artery 30%, according to the SES definition. A patient could hence have an SES value from 0 to 100. 

### 2.6. Lipidomics Analysis

Venous blood samples were collected at the four study visits in evacuated plastic tubes and centrifuged in a cooling system at 1560 g for 10 min at −40 °C and stored at −80 °C as plasma aliquots prior to lipidomic analyses. 

The order of plasma samples was randomized prior to extraction and analysis. Lipids were extracted using a modified version of the Folch procedure [17]. Ten µL of 0.9% NaCl and organic solvent (120 µL, 2:1 vol:vol chloroform:methanol) containing 2.5/mL µg of internal standard solution was added to 10 μL of each plasma sample. Samples were vortex mixed and incubated on ice for 30 min, after which they were centrifuged at 9400× *g* for 3 min at 4 °C. Sixty µL from the lower lipid layer of each sample was transferred to a glass vial with an insert, and 60 µL of chloroform:methanol 2:1, vol:vol was added to each sample. Extracted lipids from plasma samples were stored at −80 °C until analysis and were randomized before analysis. 

Chromatographic separation of 1 μL of the extracted samples was performed on an Agilent 1290 Infinity ultra-high-performance liquid chromatography system connected to an Agilent 6545 quadrupole time-of-flight mass spectrometer equipped with jet stream electrospray ionization. The extracted samples were analyzed in positive electrospray ionization polarity mode (ESI+), and MassHunter B.06.01 (Agilent Technologies, Santa Clara, CA, United States) was used for all data analysis.

Data processing was performed using open-source software MZmine 2.3.4. Details of the mass spectrometry (MS) data processing are shown in supplementary materials. Peaks were identified using a custom database search and normalized using lipid class specific internal standards and MS/MS data, as described [18]. Unknown lipids were normalized to the nearest eluting internal standard. The custom database used in this study was recently assessed as part of the NIST lipidomics ring study, comprising 31 laboratories worldwide [19]. Quality control was performed throughout the dataset by including blanks, pure standard samples, extracted standard samples, and pooled plasma samples from this study. Mean relative standard deviation for internal standards in all samples was 25.4% (raw variation), and the mean relative standard deviation percent for the identified lipids in the pooled samples (*n* = 5) was 15.6%. For the lipid species, fatty acyl chains were abbreviated Cx:y, where x represents the number of carbon atoms and y the number of double bonds of the fatty acyl chain. 

### 2.7. Statistical Analysis

Statistical analyses were performed using R v. 3.6.1. Missing values (0.03%) were imputed using random forest algorithm implemented in the R package missForest. Only identified lipids were subjected to statistical analysis. 

We applied a multilevel random forest algorithm implemented in the R package MUVR [20] to investigate effects of diet on plasma lipidome. The multilevel analysis is applied to dependent data structures in which different treatments are administered to the same subjects and has been successfully used to exploit differences related to dietary intervention in cross-over studies [20]. All variables were normalized to unit variance (z-scores). The number of components were selected based on repeated double cross-validation, and significance of multivariate models was assessed by permutation tests (*n* = 1000, Appendix A). A common baseline effect was assumed for both interventions, since no differences in plasma lipidome were observed between baseline and the end of the wash-out period (Appendix A). We further assessed the effect of VD vs. MD on each target lipid using generalized linear mixed models (R package lme4). Fixed factors included diet, sequence of diet allocation, and their interaction, with baseline value as covariate and subject as random factor. Data were log_2_-transformed before analysis, and *p* values were adjusted for multiple comparisons using the Benjamini-Hochberg false discovery rate (FDR). A value of *p* < 0.05 was considered significant. We additionally assessed differences from baseline in lipidome of each diet intervention (VD or MD).

To identify lipids with similar physiological and molecular characteristics that differed between VD and MD, a co-expression network of lipids was constructed using the weighted gene correlation network analysis (WGCNA) [21]. In brief, we calculated a correlation matrix containing all pairwise Pearson’s correlations between lipids measured after intervention, first removing outliers to improve data quality (Appendix A). We then defined a signed hybrid network, and a power of 8 was selected by the Scale-Free Topology criterion (model fitting index R^2^ > 0.8) (Appendix A). The lipids were hierarchically clustered using the distance measure, and modules were constructed by establishing a height cutoff for the resulting dendrogram using a dynamic tree-cutting algorithm, with a minimum module size of five. The resulting lipid modules were assigned color names and identified using the eigenvector of each module, designated the module eigenlipid (ME), defined as the first principal component of the standardized expression profiles, considered the best archetype of the standardized module expression data. Twelve lipids were not grouped in any module (Appendix A). Differences in ME between VD and MD were assessed by paired *t* tests. 

We further assessed whether lipids that differed between VD and MD were associated with CAD burden. First, a matrix of lipids that differed between VD and MD was calculated, normalized, and entered into a principal component analysis to discern patterns and account for inter-correlations. We determined the number of components according to the criteria of a very simple structure. Associations between PCA scores based on lipidome data and CAD burden were analyzed using linear regression adjusted for age, sex, and BMI measured at baseline. We investigated associations of plasma lipids measured at baseline with CAD burden, using linear regression. In addition, we computed Spearman correlation coefficients between lipid patterns and the clinical factors that differed between VD and MD, including BMI, oxidized LDL-C, LDL-C, and TC. 

## 3. Results

### 3.1. Characteristics of Participants

A total of 150 patients were assessed for eligibility, and 31 that agreed to participate in the study were randomized 1:1 to either VD/MD or MD/VD. During the first intervention period one participant from each diet group withdrew, and, during the second intervention period, a further two participants withdrew, also one from each group. Thus, 27 completed both intervention periods. Baseline characteristics of the study population in the VERDI trial are shown in Table 1. The majority of study participants were male (*n* = 24) with a median age of 67 years and a median BMI of 27.5 kg m^2^. Before enrolment, 24 (77%) patients had experienced myocardial infarction, and eight (26%) had been diagnosed with stable or instable angina. All study participants received statin therapy, and 94% took aspirin. 

The baseline characteristics of the participants relative to number of stenotic lesions in the coronary arteries and SES are shown in Supplemental Table 1. The SES and the number of stenotic lesions in the coronary arteries were strongly correlated (r = 0.80, *p* < 0.001), and also correlated with systolic blood pressure (r = 0.34, *p* < 0.05), BMI (r = -0.49, *p* < 0.001), and weight (r = -0.41, *p* < 0.001). Subjects with ≥4 stenotic lesions compared to those with <4 stenotic lesions had significantly lower Hba1c levels. Individuals with SES > 18 were older and had lower weight than those with SES ≤ 18. 

### 3.2. Plasma Lipidome

In total, 214 lipids were identified, including 81 glycerolipids (80 triacylglycerols (TG) and 1 diacylglyceride (DAG)); 95 glycerophospholipids (55 phosphatidylcholine (PC), 2 phosphatidylglycerols (PG), 3 phosphatidylinositols (PI), 7 phosphatidylethanolamines (PE), 19 alkylphosphatidylcholines (O-PC), 7 lyso-phosphatidylcholines (LPC)); 33 sphingolipid (22 sphingomyelins (SM), 8 ceramides (Cer), 2 hexosylceramides (HexCer), 1 lactosylceramides (LacCer)) and 7 sterol lipid classes (7 cholesteryl esters (CE)) (Appendix A).

The plasma lipid profiles differed significantly with diet, with a misclassification rate of 96% (Appendix A). Plasma levels of 41 lipids differed significantly in VD and MD (Figure 3), including 18 TGs, 12 O-PCs, 6 PCs, 2 PEs, LPC (16:0), CE (18:0), and Cer (d18:1/16:0) when assessed using univariate statistics (FDR *p* < 0.05, Appendix A). Specifically, compared with the MD, the VD resulted in higher levels of 11 TGs and lower levels of 7 TGs, 9 O-PCs, 5 PCs, two PEs, CE (18:0), Cer (d18:1/16:0), and LPC (16:0). Plasma levels of 64 lipids significantly differed from baseline after VD, while 11 differed from baseline with MD (FDR *p* < 0.05, Appendix A, Appendix A). Lipids showing greatest difference between diets were TGs, PCs, and O-PCs.

We further clustered lipids into co-expression modules to identify those with similar physiological and molecular characteristics and reduce the dimensions of the lipidomics data, clustering 202 of 214 lipids into 12 modules. Each module was, in general, dominated by lipids of the same class (Figure 4, Appendix A). Compared with MD, the VD resulted in lower level of eigenlipid modules dominated by PCs (MEyellow and MEturquoise), SMs (MEturquoise), and TGs with a median of one unsaturated carbon/carbon bond (MEmagenta) and a greater level of eigenlipid modules including large numbers of long-chain TGs with polyunsaturated fatty acyls (Figure 4).

Of note, WGCNA-modules that differed in VD and MD were significantly correlated with cardiovascular risk factors that were lower after VD than after MD (Figure 4, Appendix A). Specifically, the module of SMs (MEturquoise) was positively correlated with oxidized LDL-C, LDL-C, and TC, while it was inversely correlated with BMI and bodyweight. The module of PCs (MEyellow) strongly correlated with TC. The module mainly comprised of ceramides (MEblack), showed strongest correlation with oxidized LDL-C (r = 0.37, *p* = 0.007) compared to other modules, but did not differ between VD and MD. 

### 3.3. Associations between Plasma Lipids and Coronary Artery Disease Burden

Principle components analysis was applied on 74 lipids that differed between VD and MD (*p* < 0.05, Appendix A) to account for the inter-correlations among lipids (correlation range −0.51 to 0.85). Four components accounted for 64% of the total variance in lipid levels (Table 2). We observed an inverse association of the SES with the lipidome PCA component 2, representing long-chain polyunsaturated TGs, which were higher after the VD than after MD. The PCA component 4, comprising PCs that were reduced in VD compared with MD, was positively associated with SES and the total number of stenotic lesions. Significant correlations of the PCA components and the certain characterized lipid species with VD-induced reduction of LDL-C and TC were also observed (Table 2, Appendix A). 

In addition, we found baseline levels of 10 lipids that were associated with the total number of stenotic lesions and six associated with the SES (Appendix A). Among these, PE (O-38:5), PC (36:1), PC (38:1), and PC (39:6) were associated with both values. No association remained significant after correction for multiple testing. 

## 4. Discussions

In this crossover randomized dietary intervention study in CAD patients on standard medical therapy, a four-week VD showed significant impact on plasma lipids, in particular TGs, PCs, O-PCs, and SMs, compared to an isocaloric MD. Lipid clusters dominated by TGs and PCs were positively correlated with VD-induced improvements in cardiometabolic risk factors. We found that favorable changes in long-chain polyunsaturated containing TGs induced by VD were inversely associated with CAD burden, which has not been previously reported. Our findings indicate that the VD led to alteration in level of several plasma lipotoxic lipids associated with subclinical coronary artery disease and development of recurrent coronary artery events.

### 4.1. Diet Effects on Plasma Lipidome

An elevated level of fasting blood triglycerides has been considered an independent risk factor for CAD [22], but findings of randomized controlled trials regarding the effects of plant-based diet on blood triglycerides are inconsistent [4,6,7]. Our initial analysis of data of the current trial did not show difference in total TG between VD and MD [14]. To our knowledge, no previous study has been reported where the impact of a vegetarian diet on the quality of triglycerides and its proportions of saturated, monounsaturated, and polyunsaturated fatty acyl content was evaluated. Our data using high-throughput lipidomics clearly showed the VD to result in higher levels of long-chain and polyunsaturated TGs and lower levels of lipotoxic lipids, such as TGs with saturated fatty acyl chains as well as and specific species of ceramides than observed with MD. In a primary preventive cohort of 685 healthy individuals, short-chain saturated fatty acyl chains of TGs were found to be associated with higher risk of cardiovascular diseases development [23]. It has also been reported that CAD patients exhibited lower levels of unsaturated TGs in the epicardial adipose tissue than people without CAD [24]. We found TGs containing linoleic acid to be higher after the VD than after MD. Studies have shown inverse associations of high levels of circulating phospholipids containing linoleic acid with CAD mortality [25] and subclinical CAD [26,27]. 

Our findings highlight the need to address the effect of TG species separately, as opposed to the total blood triglyceride concentration. The measure of total TGs, rather than the assessment of individual components as conducted in the present study, may explain inconsistent conclusions regarding effects of vegetarianism on blood triglycerides in the literature. 

We found that diet interventions affected levels of several glycerophospholipids, in particular PCs, PEs, LPCs, and SMs. These lipid species are the main constituents of cell membranes and are also involved in signal transduction and various physiological functions and have been associated with development of CAD and its risk factors [28,29]. Compared with the MD, the four-week VD was associated with lower levels of PCs and SMs containing fatty acyl chains C38:5, C38:6, C32:0, C16:0, and C24:1, which are associated with CAD risk and cardiovascular mortality [23,28,29]. The VD also resulted in lower levels of several alkylphospholipids (O-PC) species than MD and compared to baseline. Alkylphospholipids have been reported to be inversely associated with multiple metabolic diseases [30,31,32]. Meanwhile, Meikle et al. [33] observed that postprandial levels of ether-linked phospholipids increased after a dairy meal but decreased after a soy-based meal. Further studies are required to elucidate whether modulation of alkyl-phosphatidylcholine levels by drugs or diet could have beneficial effects on CAD.

Ceramides are known to affect several metabolic processes linked to CAD, including LDL aggregation and uptake, endothelial dysfunction, and multiple inflammatory processes [10,34]. The ceramide species Cer(d18:1/16:0), Cer(d18:1/18:0), Cer(d18:1/24:1), and Cer(d18:1/24:0) have been shown predictive of cardiovascular events, especially cardiovascular death, independent of traditional risk factors [10,11]. Assessment of ceramide biomarkers has been introduced into clinical practice at the Mayo clinics to assess risk of CAD events [35]. We observed lower Cer(d18:1/16:0) levels with the VD than with MD. In addition, lower levels of Cer(d18:1/16:0), Cer(d18:1/22:0), HexCer(d18:1/22:0), LacCer(d18:1/16:0) were observed after VD than at baseline, partly supporting its benefits in management of cardiovascular disease. 

The VD improved the plasma lipid profile predominately by reducing levels of lipotoxic lipid species associated with adverse cardiometabolic health outcomes such as TGs with saturated fatty acyls and certain species of glycerophospholipids, and ceramides. The VD caused higher levels of long-chain polyunsaturated TGs that have been reported associated with beneficial cardiometabolic outcomes.

### 4.2. Associations of Plasma Lipids with Coronary Artery Disease Burden

We found lipid species that differed between VD and MD to be associated with CAD burden, as well as with CAD risk factors improved by VD, including BMI, oxidized LDL-C, LDL-C, and TC [14]. These results are in agreement with previous studies showing an association of elevated lipoprotein(a) and LDL-C with CAD burden as assessed by the SES [36], and other coronary risk scores [37,38]. Importantly, we found that long-chain polyunsaturated TGs, which were higher after VD compared with MD, to be inversely associated with SES in CAD patients. Several PCs that were lower after VD than after MD were associated with a greater CAD burden as assessed by the total number of stenotic lesions. Our findings strengthen the hypothesis of benefits of VD for prevention of CAD events via modulation of the lipid profile. 

Participants with a higher SES had lower body weight than those with lower scores. Previous studies have indicated that overweight patients undergoing PCI have better clinical outcomes compared to lean or normal weight patients [39]. However, this may be a result of reverse causation and confounding by smoking [40]. We found that participants with a higher SES were more likely to have a history of myocardial infarction than stable/unstable angina, suggesting that lower weight in this group of patients might have been due to reverse causality. 

### 4.3. Strengths and Limitations

The strengths of the reported study include its cross-over design, well-characterized subjects receiving standard medical therapy, and a high rate of study completion. The applied untargeted lipidomics covered a wide range of lipid species, enabling us to investigate distinct impacts of diet intervention on different lipid species. Moreover, the present cross-over study design reduces the likelihood of bias and we estimated that any potential confounders (e.g., estimated glomerular filtration rate indicating renal function, diabetic status, statin or ezetimibe treatment) will be equally distributed in each sequence (MD-VD vs. VD-MD). Further strengths are comprehensive statistical analyses, including both traditional and machine learning methods including multivariate statistics (supervised multilevel random forest, WGCNA network analysis, and unsupervised principal component analysis) and univariate analysis, allowing identification of lipid clusters associated with diet intervention, cardiometabolic risk factors, and CAD burden. Of note, by using multilevel modelling particular designed to analyze dependent data structures in which different treatments are administered to the same subjects, we could successfully exploit within-individual variation in lipidome related to dietary intervention.

Limitations include small study size and the few women subjects which may decrease generalizability. The lipidomics approach applied in this study cannot provide absolute concentrations of lipid species in plasma. Although there are constant advances in lipid analysis, complete lipidome analysis remains challenging. Our study included relative concentrations of 214 lipids in five classes. However, we cannot exclude the possibility that our lipidomics approach may have failed to detect other lipids that could be affected by VD. Atherosclerotic burden was visually assessed without absolute quantification and with no invasive versus noninvasive evaluation. However, the two scores for assessing the atherosclerotic burden were highly correlated and consistent results were obtained regarding associations between plasma lipids and scores, supporting that the quantitative measurements were adequate. Although studies repeatedly showed significant protective effects of a vegetarian diet on CVD prevention [41,42,43], vegetarian diets may cause mineral deficiencies, e.g., iron, calcium, and zinc [44]. Previous studies have shown the associations between abnormal mineral metabolism and high risk of CVD [45,46], particularly in patients with chronic kidney disease. However, such health risks of vegetarian diets greatly depend on the types and amounts of foods actually consumed, which should be further explored. Additionally, the dietary interventions were relatively short (two months) and we did not have any data on long-term results (i.e., if any of the subjects turned vegetarian), which is of great interest.

## 5. Conclusions 

Our exploratory analysis of the randomized, cross-over study revealed that a VD compared with a diet including daily meat consumption improved plasma lipid profiles, in particular TGs, PCs, O-PCs, and SMs in CAD patients on standard medical therapy. The levels of plasma lipotoxic species (PCs, O-PCs, and saturated TGs), associated with increased risk of metabolic disease, were lower after a VD compared with MD. Contrastingly, VD increased TGs with long-chain polyunsaturated fatty acyls that were found to be inversely associated with CAD burden. Our results support the hypothesis that a VD may be beneficial for secondary prevention of CAD via modulation of lipid profiles. Future studies of larger size are warranted to confirm the findings of alterations of lipid profiles by a vegetarian diet. 

## Figures and Tables

**Figure 1 nutrients-12-03586-f001:**
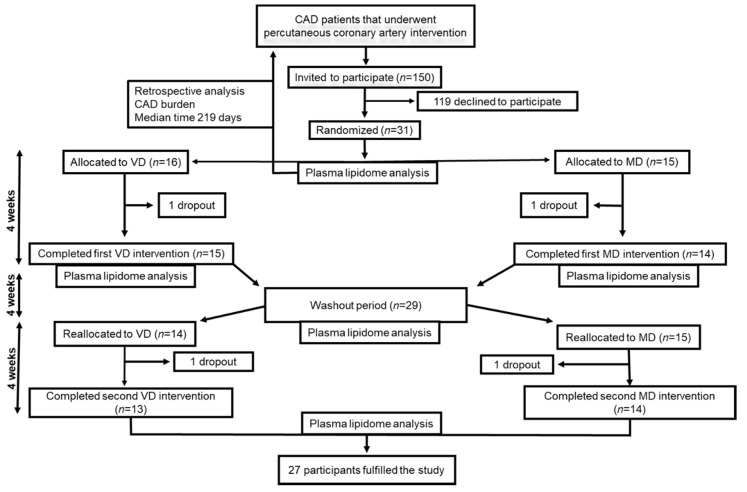
The workflow of study. CAD, coronary artery disease; MD, meat diet; VD, vegetarian diet.

**Figure 2 nutrients-12-03586-f002:**
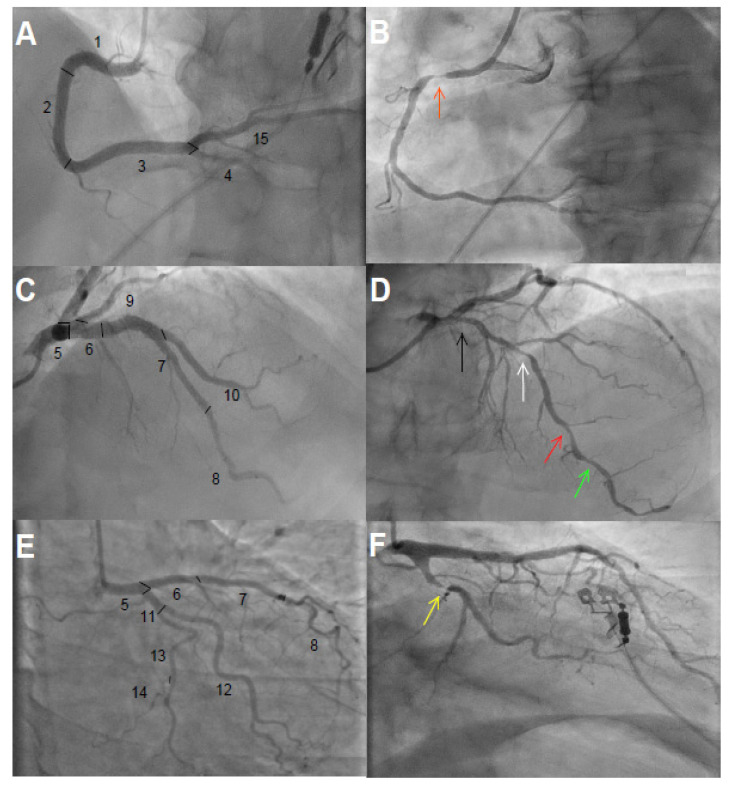
Coronary angiograms. (**A**,**C**,**E**) show 15 segments of the coronary tree classified according to the American Heart Association without significant obstructive coronary artery disease. (**A**): 1 = Proximal right coronary artery, 2 = middle right coronary artery, 3 = distal right coronary artery, 4 = right posterior descending artery, and 15 = left posterior descending artery; (**C**): 5 = left main coronary artery, 6 = proximal left anterior descending artery, 7 = middle left anterior descending artery, 8 = distal left anterior descending artery, 9–10 = first and diagonal branches of left anterior descending artery; (**E**): 11 = proximal left circumflex, 12 = obtuse marginal branch, 13 = distal left circumflex, 14 = posterolateral branch. (**B**): significant obstructive coronary artery disease with stenosis ~90% (orange arrow) in segment 1; the longitudinal extent of atherosclerosis was ~70%. (**D**): Arrows from right to left show ~40% (black), ~60% (white), ~60% (red), and ~40% (green) stenosis in segments 6, 7, 8, and 8, respectively. Longitudinal extent of atherosclerosis was ~80% in segment 6, ~90% in segment 7, and ~80% in segment 8. (**F**): ~90% stenosis in segment 11 (yellow arrow) with estimated ~50% longitudinal atherosclerosis.

**Figure 3 nutrients-12-03586-f003:**
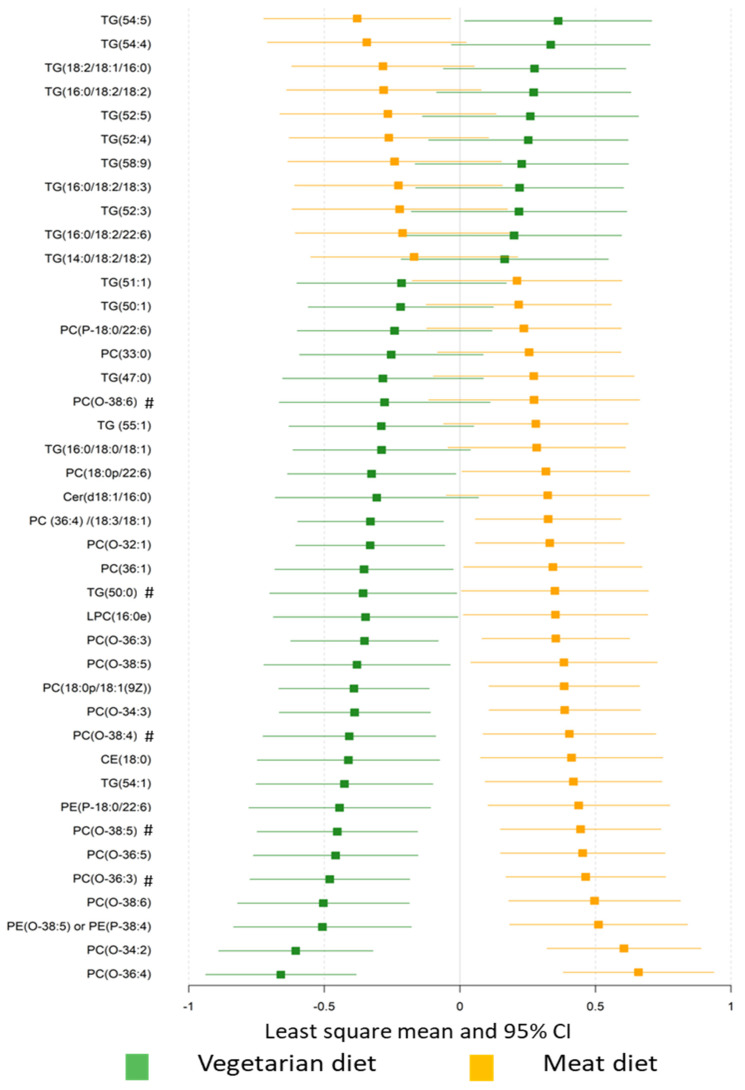
The effect of diet on plasma lipid profiles. Plasma lipids significantly differed between the vegetarian and meat diet by generalized linear mixed modelling (FDR *p* < 0.05). Standardized values of least-squares mean with 95% confidence intervals are presented for comparison. TG, triacylglycerols; DAG, diacylglyceride; PC, phosphatidylcholine; PG, phosphatidylglycerol; PI, phosphatidylinositol; PE, phosphatidylethanolamine; O-PC, alkylphosphatidylcholines; LPC, lyso-phosphatidylcholine; SM, sphingomyelin; Cer, ceramides; CE, cholesteryl ester. # Lipids with same fatty acyl chains (Cx:y) were presented in the dataset, where x represents the number of carbon atoms and y the number of double bonds of the fatty acyl chain.

**Figure 4 nutrients-12-03586-f004:**
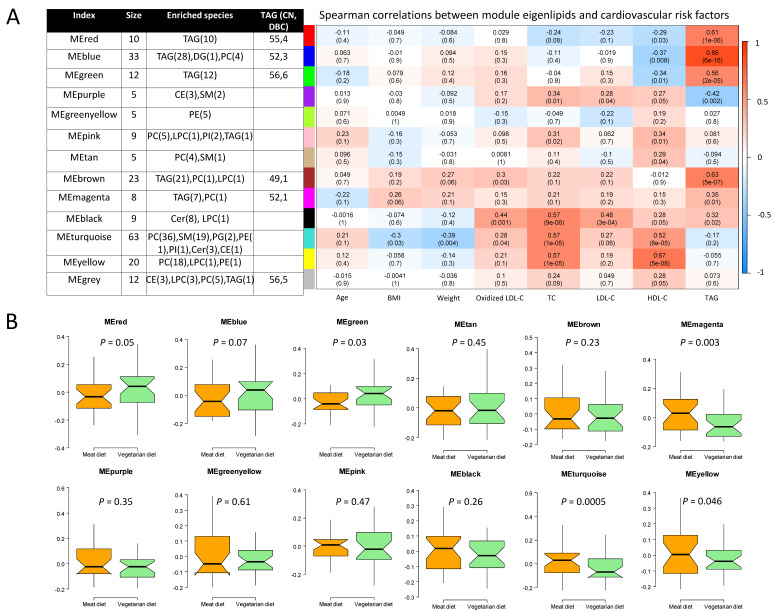
Co-expression modules of lipids determined by the weighted gene correlation network analysis. Module size, lipid species, and range of total number of carbons (CN) and number of double bonds (DBC) of TG (**A**, right), Spearman’s correlation coefficient between modules and cardiovascular risk factors (**A**, left), and difference in eigenlipid of modules between VD and MD (**B**). Cer, ceramide; CE, cholesteryl ester; DAG, diacylglyceride; HDL-C, High-density lipoprotein cholesterol; HexCer, hexosylceramide; LacCer, lactosylceramide; LDL-C, low-density lipoprotein cholesterol; LPC, lyso-phosphatidylcholine; O-PC, alkylphosphatidylcholines; PC, phosphatidylcholine; PE, phosphatidylethanolamine; PG, phosphatidylglycerol, PI, phosphatidylinositol; SM, sphingomyelin; TG, triglyceride.

**Table 1 nutrients-12-03586-t001:** Baseline characteristics of VERDI study participants.

Characteristics	All (*n* = 31)	VD (*n* = 16)	MD (*n* = 15)
Age years, median(range)	67 (63–70)	67 (65–70)	68 (61–70)
Sex, male, *n* (%)	29 (94%)	15 (94%)	14 (93%)
Myocardial infarction, *n* (%)	24 (77%)	6 (63%)	6 (93%)
Angina, *n* (%)	8 (26%)	7 (44%)	1 (7%)
Diabetes mellitus type 2, *n* (%)	2 (7%)	2 (13%)	0 (0%)
Hypertension, *n* (%)	17 (55%)	10 (63%)	7 (47%)
BMI, kg/m^2^	28 ± 2.9	28 ± 3.3	27 ± 2.5
Systolic Bp, mm Hg	139 ± 17.4	140 ± 17.4	138 ± 18.0
Diastolic Bp, mm Hg	87 ± 9.6	88 ± 10.6	87 ± 8.7
Total cholesterol, mmol/L	3.5 ± 0.6	3.5 ± 0.73	3.4 ± 0.44
LDL-C, mmol/L	1.6 ± 0.4	1.6 ± 0.5	1.6 ± 0.4
HDL-C, mmol/L	1.3 ± 0.3	1.3 ± 0.4	1.2 ± 0.2
Triacylglycerol, mmol/L	1.1 ± 0.3	1.1 ± 0.4	1.1 ± 0.3
eGFR, mL/min per 1.73 m^2^	76.4 ± 9.7	75.1 ± 7.6	77.7 ± 11.7
HbA_1c_ mmol/mol, median (range)	39 (36–40)	39 (36–42)	39 (36–40)
Hs-crp mg/L, median (range)	0.7 (0.5–1.7)	0.8 (0.4–1.7)	0.7 (0.4–1.7)
Statin, *n* (%)	31 (100%)	16 (100%)	15 (100%)
Ezetimibe, *n* (%)	7 (23%)	4 (25%)	3 (20%)
ASA, *n* (%)	29 (94%)	15 (94%)	14 (93%)
P2y_12_ inhibitors, *n* (%)	20 (65%)	8 (50%)	12 (80%)
Beta-blockers, *n* (%)	28 (90%)	14 (88%)	14 (93%)
ACE inhibitors/ARBs, *n* (%)	27 (87%)	13 (81%)	14 (93%)
CCB, n (%)	11 (36%)	6 (38%)	5 (33%)

Data are presented as median [interquartile range], *n* (%) or mean ± standard deviation. ACE inhibitors/ARB, angiotensin converting enzyme inhibitors or angiotensin II receptor blockers; ASA, acetylsalicylic acid; Bp, blood pressure; CCB, calcium channel blockers; eGFR, estimated glomerular filtration rate; HbA1c, glycated hemoglobin; HDL-C, high-density lipoprotein cholesterol; Hs-CRP, high-sensitivity c-reactive protein; LDL-C, low-density lipoprotein cholesterol; mmHg, mm of mercury; P2Y12 inhibitors, clopidogrel or ticagrelor.

**Table 2 nutrients-12-03586-t002:** Associations between of orthogonal lipids pattern described by principal components derived from a principal component analysis with coronary artery disease burden and cardiometabolic risk factors.

	PCA Component 1	PCA Component 2	PCA Component 3	PCA Component 4
Associations with CAD burden (β-coefficient ± SE)				
Sullivan extent score	0.05 ± 0.06	−0.25 ± 0.10 *	−0.24 ± 0.23	0.87 ± 0.51 *
Number of stenotic lesions	−0.005 ± 0.02	−0.02 ± 0.04	−0.008 ± 0.08	0.31 ± 0.16 *
Association with cardiometabolic risk factors that were improved after VD				
Oxidized LDL-C	0.08	−0.14	0.45 *	−0.12
LDL-C	0.02	−0.60 *	0.30 *	−0.13
TC	0.09	−0.59 *	0.31 *	−0.08
BMI	0.31 *	0.16	0.21	−0.07

Associations of baseline lipids with the Sullivan extent score (SES) and the total number of stenotic lesions were assessed using linear regression adjusted for age, sex, BMI, and total cholesterol. The β-coefficient and standard error obtained from linear regression are presented. * denotes significant association (*p* < 0.05).

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
