# Peer review of "Effects of a Lacto-Ovo-Vegetarian Diet on the Plasma Lipidome and Its Association with Atherosclerotic Burden in Patients with Coronary Artery Disease—A Randomized, Open-Label, Cross-over Study"

_nutrients, 2020, doi:10.3390/nu12113586_

Round 1
Reviewer 1 Report
The manuscript is very interesting and current, considering the high incidence of cardiovascular morbidity and mortality, although it was already studied in health subjects.
Josefine Nebl, Sven Haufe, Julian Eigendorf, Paulina Wasserfurth, Uwe Tegtbur, Andreas Hahn. Exercise capacity of vegan, lacto-ovo-vegetarian and omnivorous recreational runners. J Int Soc Sports Nutr. 2019; 16: 23. Published online 2019 May 20.
Moreover the authors have already reported the similar results with the same patients examined.
Demir Djekic, Lin Shi, Harald Brolin, Frida Carlsson, Charlotte Särnqvist, Otto Savolainen, Yang Cao, Fredrik Bäckhed, Valentina Tremaroli, Rikard Landberg, and Ole Frøbert.Effects of a Vegetarian Diet on Cardiometabolic Risk Factors, Gut Microbiota, and Plasma Metabolome in Subjects With Ischemic Heart Disease: A Randomized, Crossover Study. Originally published6 Sep 2020https://doi.org/10.1161/JAHA.120.016518Journal of the American Heart Association. 2020;9
The authors should be report the eGFR of patients, considering its influence on lipidome and on cardiovascular risk.
It would have been interesting to know the alterations of mineral metabolism, considering their influence on cardiac and vascular calcifications and their contribution in determining a high cardiovascular risk, particularly in CKD patients.
Author Response
The manuscript is very interesting and current, considering the high incidence of cardiovascular morbidity and mortality, although it was already studied in health subjects.
Josefine Nebl, Sven Haufe, Julian Eigendorf, Paulina Wasserfurth, Uwe Tegtbur, Andreas Hahn. Exercise capacity of vegan, lacto-ovo-vegetarian and omnivorous recreational runners. J Int Soc Sports Nutr. 2019; 16: 23. Published online 2019 May 20.
Moreover the authors have already reported the similar results with the same patients examined.
Demir Djekic, Lin Shi, Harald Brolin, Frida Carlsson, Charlotte Särnqvist, Otto Savolainen, Yang Cao, Fredrik Bäckhed, Valentina Tremaroli, Rikard Landberg, and Ole Frøbert. Effects of a Vegetarian Diet on Cardiometabolic Risk Factors, Gut Microbiota, and Plasma Metabolome in Subjects With Ischemic Heart Disease: A Randomized, Crossover Study. Originally published6 Sep 2020https://doi.org/10.1161/JAHA.120.016518 Journal of the American Heart Association. 2020;9
Response: We thank the reviewer for these remarks. The reviewer reported that a study has already been conducted in healthy subjects (Nebl et al, J Int Soc Sports Nutr. 2019; 16: 23). However, the cited study investigated the effects of vegan and vegetarian diets on the exercise capacity in runners rather than the effect of a vegetarian diet on plasma lipids (lipidome), which was conducted in the present study. The reviewer correctly states that similar results has been published in the same patients (Djekic et al, JAHA). In our previously published study, we investigated the effects of a vegetarian diet on the plasma metabolome. However, our new study addresses lipids and we investigated the effects on vegetarian diet on plasma lipidome, and these findings are novel. We found that the vegetarian diet increased levels of 11 triacylglycerols and lowered levels of 7 triacylglycerols, glycerophospholipids, cholesteryl ester (18:0) and ceramide (d18:1/16:0) compared with meat diet. The vegetarian diet led to improved levels of several lipotoxic lipids, previously associated with increased risk of recurrent coronary artery disease events (Stegemann C, et al. Circulation 2014; 129: 1821-1831. 2014/03/14; Hilvo M, et al. Eur Heart J 2019 2019/06/19). In that context, we think the results are of interest and improves our understanding of the underlying mechanisms which may benefit coronary artery disease patients through lifestyle/diet modifications.
Reviewers comment 1: The authors should be report the eGFR of patients, considering its influence on lipidome and on cardiovascular risk.
Response: We thank the reviewer for this comment. We agree that renal function/failure may influence the plasma lipidome. We have now provided eGFR values of patients at baseline in the Table 1. Although a slight difference was found for eGFR between VD (75.1±7.6 mL/min per 1.73 m2) and MD (77.7±11.7 mL/min per 1.73 m2) at baseline, we believe that it would not influence the results regarding effects of vegetarian diet on the plasma lipidome, due to the cross-over study design. The present study design reduces the likelihood of bias and we estimate that any potential confounder will be equally distributed in each sequence (MD- VD vs VD-MD). By using multilevel modelling designed to analyze dependent data structures in which different treatments are administered to the same subjects, we could successfully exploit within-individual variation in lipidome related to dietary intervention.
This has now been clarified on p. 14 and line. 410-413, and 417-420, as shown below.
“Moreover, the present cross-over study design reduces the likelihood of bias and we estimated that any potential confounders (e.g. estimated glomerular filtration rate indicating renal function, diabetic status, statin or ezetimibe treatment) will be equally distributed in each sequence (MD- VD vs VD-MD). Further strengths are comprehensive statistical analyses, including both traditional and machine learning methods including multivariate statistics (supervised multilevel random forest, WGCNA network analysis, and unsupervised principal component analysis) and univariate analysis, allowing identification of lipid clusters associated with diet intervention, cardiometabolic risk factors, and CAD burden. Of note, by using multilevel modelling designed to analyze dependent data structures in which different treatments are administered to the same subjects, we could successfully exploit within-individual variation in lipidome related to each dietary intervention.”
Reviewers comment 2: It would have been interesting to know the alterations of mineral metabolism, considering their influence on cardiac and vascular calcifications and their contribution in determining a high cardiovascular risk, particularly in CKD patients.
Response: We thank the reviewer’s suggestion and we agree that the effects of a vegetarian diet on mineral metabolism would be interesting to study. However, we believe that it is a bit outside the focus of this paper and we have not measured any lipids related to the mineral metabolism. We have now considered this as a limitation of our study and have briefly discussed potential effects of vegetarian diet on mineral metabolism at page 15, line 441-446, as shown below.
“Although studies repeatedly showed significant protective effects of a vegetarian diet on CVD prevention 1-3, vegetarian diets may cause mineral deficiencies, e.g. iron, calcium, and zinc 4. Previous studies have shown the associations between abnormal mineral metabolism and high risk of CVD 5, 6, particularly in patients with chronic kidney disease. However, such health risks of vegetarian diets greatly depend on the types and amounts of foods actually consumed, which should be further explored.”
References
- Dinu M, Abbate R, Gensini GF, et al. Vegetarian, vegan diets and multiple health outcomes: A systematic review with meta-analysis of observational studies. Critical Reviews in Food Science and Nutrition 2017; 57: 3640-3649. DOI: 10.1080/10408398.2016.1138447.
- H K, S L and ND B. - Vegetarian Dietary Patterns and Cardiovascular Disease. D - 0376442: - 54-61.
- M DA. - Vegetarian diets in cardiovascular prevention. D - 9815942: - 735-745.
- JP R, J L, B P, et al. - Multiple Health Benefits and Minimal Risks Associated with Vegetarian Diets. D - 101587480: - 374-381.
- C V, N A, C M, et al. The interplay between mineral metabolism, vascular calcification and inflammation in Chronic Kidney Disease (CKD): challenging old concepts with new facts. D - 101508617: - 4274-4299.
- A G, S M, G C, et al. - Fibroblast Growth Factor 23: Mineral Metabolism and Beyond. D - 7513582: - 83-95.
Reviewer 2 Report
The authors analyzed the effects of a lacto-ovo-vegetarian diet on the plasma lipidome and its association with atherosclerotic burden in patients with coronary artery disease. Presented study had a relatively small number of patients, despite sample size calculation. The manuscript is well written, with potentially interesting topic adding meaningful information to current knowledge. a few more issues should be considered:
- how many patients uderwent CABG? Some studies report better compliance with patietns underging more severe procedure with more visible extent as compared to PCI. Those patients might be more awere of secondary prevention, furthermore baseline adherence to recommendations in this group might be higher
- any information of statin dose and type?Have authors evaulated kidney and liver function? some statins are mainly metabolised by liver or kidneys, thus serum concentration might be different in those patients with potential impact on outcome.
- there was difference in diabetes mellitus rate. DM might significalnty impact level of triglycerides, thus potential impact of outcome can not be ruled out.
- any data for long-term results and impact on rate of MACCE?
Author Response
Reviewer #2:
The authors analyzed the effects of a lacto-ovo-vegetarian diet on the plasma lipidome and its association with atherosclerotic burden in patients with coronary artery disease. Presented study had a relatively small number of patients, despite sample size calculation. The manuscript is well written, with potentially interesting topic adding meaningful information to current knowledge. A few more issues should be considered:
Response: We thank the reviewer for these remarks.
Reviewers comment 1: How many patients uderwent CABG? Some studies report better compliance with patietns underging more severe procedure with more visible extent as compared to PCI. Those patients might be more awere of secondary prevention, furthermore baseline adherence to recommendations in this group might be higher.
Response: We appreciate the question raised by the reviewer. However, in our study, none of the patients underwent the coronary artery bypass grafting. All patients underwent percutaneous coronary intervention.
Reviewers comment 2: any information of statin dose and type? Have authors evaluated kidney and liver function? some statins are mainly metabolised by liver or kidneys, thus serum concentration might be different in those patients with potential impact on outcome.
Response: We thank the reviewer for the question. All patients were on statin treatment and seven patients were treated with ezetimibe. We agree that statins may influence the plasma lipidome. However, due to the cross-over study design and that no changes were made in statin dose or type or ezetimibe dose, we think that this could not have impacted the results.
This has now been clarified on p. 14 and line. 410-413, and 417-420, as shown below.
“Moreover, the present cross-over study design reduces the likelihood of bias and we estimated that any potential confounders (e.g. estimated glomerular filtration rate indicating renal function, diabetic status, statin or ezetimibe treatment) will be equally distributed in each sequence (MD- VD vs VD-MD). Further strengths are comprehensive statistical analyses, including both traditional and machine learning methods including multivariate statistics (supervised multilevel random forest, WGCNA network analysis, and unsupervised principal component analysis) and univariate analysis, allowing identification of lipid clusters associated with diet intervention, cardiometabolic risk factors, and CAD burden. Of note, by using multilevel modelling designed to analyze dependent data structures in which different treatments are administered to the same subjects, we could successfully exploit within-individual variation in lipidome related to each dietary intervention.”
Reviewers comment 3: there was difference in diabetes mellitus rate. DM might significantly impact level of triglycerides, thus potential impact of outcome cannot be ruled out.
Response: Only 2 patients had diabetes and none of the patients had anti-diabetic meditations. We agree that diabetic mellitus might impact level of triglycerides, but as we clarified above, by using a cross-over study design, we explored systematic within-individual differences in lipids related to dietary intervention.
As shown above, this has now also been clarified in the manuscript on p. 14 and line. 410-413, and 417-420.
Reviewers comment 4:- any data for long-term results and impact on rate of MACCE?
Reponses: In our study the interventions provided lasted for two months. We do not have any data on long-term results (i.e. if any of the subjects turned vegetarian), but we agree that it is of great interest. This has been added as a limitation in our manuscript, at page15, line 446-448. Moreover, there were no major events during the interventions.
“Additionally, the dietary interventions were relatively short (two months) and we did not have any data on long-term results (i.e. if any of the subjects turned vegetarian), which is of great interest.”
Additional changes to the manuscript:
- Clinical Trial Registration link has been changed and URL: https://www.clinicaltrials.gov has now been used.
- The last sentence of the abstract has been revised: The VD favorably changed levels of several lipotoxic lipids that have previously been associated with increased risk of coronary events in CAD patients.
- Abbreviation TAG was entirely removed and TG has now been used throughout the paper.
- We have moved the Table 2 ‘Correlation of PCA loading of lipids that differed between VD and MD’ to the supplemental material, referring to Supplemental Table 5, Loadings of lipids that differed between vegetarian diet and meat diet derived from principle component analysis.
References
- Dinu M, Abbate R, Gensini GF, et al. Vegetarian, vegan diets and multiple health outcomes: A systematic review with meta-analysis of observational studies. Critical Reviews in Food Science and Nutrition 2017; 57: 3640-3649. DOI: 10.1080/10408398.2016.1138447.
- H K, S L and ND B. - Vegetarian Dietary Patterns and Cardiovascular Disease. D - 0376442: - 54-61.
- M DA. - Vegetarian diets in cardiovascular prevention. D - 9815942: - 735-745.
- JP R, J L, B P, et al. - Multiple Health Benefits and Minimal Risks Associated with Vegetarian Diets. D - 101587480: - 374-381.
- C V, N A, C M, et al. The interplay between mineral metabolism, vascular calcification and inflammation in Chronic Kidney Disease (CKD): challenging old concepts with new facts. D - 101508617: - 4274-4299.
- A G, S M, G C, et al. - Fibroblast Growth Factor 23: Mineral Metabolism and Beyond. D - 7513582: - 83-95.
Round 2
Reviewer 1 Report
the authors reported the requested changes.
This manuscript is a resubmission of an earlier submission. The following is a list of the peer review reports and author responses from that submission.